# Shear Viscosity Overshoots in Polymer Modified Asphalts

**DOI:** 10.3390/ma15217551

**Published:** 2022-10-27

**Authors:** Martin Jasso, Giovanni Polacco, Ludovit Zanzotto

**Affiliations:** 1Chairholder, Bituminous Materials Chair, Schulich School of Engineering, University of Calgary, Calgary, AB T2N 1N4, Canada; 2Department of Civil and Industrial Engineering, University of Pisa, 56122 Pisa, Italy; 3Schulich School of Engineering, University of Calgary, Calgary, AB T2N 1N4, Canada

**Keywords:** polymer modified asphalt, styrene-butadiene-styrene (SBS), crosslinking, paving mixes, shear thickening, shear viscosity

## Abstract

Polymer modification is one of the most common methods for improving the performance of asphalt binders. Despite in-depth research, the structural modifications induced by polymers are still not well understood. In this work, steady shear viscosity measurements and cryo-scanning electron microscopy (cryo-SEM) were used to better understand the internal structure of asphalts modified by styrene-butadiene-styrene with and without sulfur as a crosslinking agent, asphalts modified by polyphosphoric acid (PPA), and quaternary asphalt blends modified by SBS, sulfur, and PPA. The results showed that polymer and asphaltenes collaborate, thus SBS forms a three-dimensional network strengthened by asphaltenes clusters. The strength, extension, and physical nature of such a network is revealed by the appearance of overshoots in the viscosity curves. Moreover, the indirect information deduced from the magnitude and shape of the shear viscosity curves successfully correlated with direct observations of the internal structure by cryo-SEM. Steady shear viscosity is thus recommended as a useful tool in studying the structural development of asphalts modified by different technologies.

## 1. Introduction

Asphalt pavements are subjected to thermo-mechanical stresses that induce failures such as rutting, fatigue, and thermal cracking. One strategy widely used to improve their performance and prolong service life is blending asphalt with polymers (polymer modified asphalts, PMA). Despite the wide number of available and tested polymers, only a few have been satisfactorily used in pavement, and the preferred ones belong to the category of thermo-plastic elastomers, like the poly (styrene-*b*-butadiene-*b*-styrene) block copolymer (SBS), whose chains are composed of alternating soft and rigid segments. The central butadiene block has high flexibility and glass transition well below the operating conditions of asphalt pavement, even in cold climates. In contrast, the external styrene blocks derived from a homopolymer have a glass transition temperature of around 100 °C.

The two types of blocks have a scarce compatibility, which results in a biphasic configuration composed of rigid polystyrene domains dispersed in a soft polybutadiene matrix. If two styrene blocks of the same chain belong to different domains, then the central butadiene segment is a bridge between those domains. The whole structure is thus a three-dimensional elastomeric network, wherein the rigid polystyrene domains constitute the nodes, while high elongation is guaranteed by the deformation of the polybutadiene segments. In these block copolymers, the rigid polystyrene domains may be arranged in several ways, with order–disorder transitions that can be induced by temperature, solvent concentration (in the case of solutions), or an applied shear stress. In the latter case, the shift from order to disorder corresponds to a shear thinning in the viscosity function [1,2].

When the polymer, in the form of a powder or pellets, is mixed with asphalt, the more compatible maltene molecules migrate into the soft polybutadiene matrix, the network swells, and the dispersed polymer droplets increase in volume. When the polymer content is high enough, this “polymer-rich” phase may become a continuous one (phase inversion), and the polymer confers its elastomeric properties to the whole material even if present in limited quantity. Whether the phase inversion takes place or not depends on the quantity and structure (linear or star shaped) of the added polymer, its compatibility with asphalt, and the operating conditions adopted during the preparation of PMAs [3]. Since the asphalt components mainly interact with the soft polybutadiene matrix, the original polymer network survives the blending procedure, and the polystyrene blocks retain their ability to organize in ordered structures [4,5,6,7,8,9,10]. The final configuration is therefore a very complex material that has physical and mechanical properties that have been extensively investigated, especially using rheology. However, despite the in-depth research, one aspect that has received little attention is the possible appearance of a shear overshoot in shear stress experiments [11,12,13]. This shear thickening is due to rearrangements of the internal structure induced by an applied stress and, even if not common in asphalt literature, it has been observed and described in several systems that can be contextualized as a subgroup of the materials, referred to as “soft matter”. These include polymer solutions [14], colloids [15], and suspensions of rigid particles [13,16]. As a general rule, this phenomenon may appear in the presence of dissolved or suspended molecules/micelles/particles showing an associative behaviour [17,18,19].

Interactions among long molecules (oligomers or polymers) or colloidal particles may lead to the formation of complex structures, such as networks or clusters, that limit the overall mobility of the fluid [20,21]. Since the networks are based on physical bonds, their stability depends on operating parameters like temperature, concentration, or flow conditions, and they have a reversible nature, so that they are referred to as transient or temporary networks.

There are many polymers that may show associating abilities when dissolved in a low molecular weight medium. If the latter is water, then it would be a water-soluble macromolecule with a limited number of hydrophobic substituents. The simplest example of an associating polymer is the telechelic one, which has an oligomeric nature and substituents located on both ends of the chain [22,23,24]. At low polymer concentrations, the hydrophobic functionalities aggregate and form the core of a flower-like micelle that has a hydrophilic part exposed to the water medium. Above a critical concentration, the micelles are close enough to form the nodes of a network. Since the physical bonds among the end caps are weak, the detachment of an end from the hydrophobic core of the flower, with a subsequent jump to form a bridge between two micelles, can be induced by an applied shear [25,26]. This effect favours the formation of an interconnected network, responsible for the shear thickening, among the nodes; this network is composed of telechelic chains, and the ends of these chains belong to neighbouring micelles [27,28,29]. Due to the weak nature of the bridges, excessive stress causes disruptions of the network, with subsequent shear thinning.

The above-described mechanism is applicable to relatively diluted solutions and has a non-reversible nature. In contrast, reversible shear thickening has been observed in several colloidal or micellar suspensions [30,31,32], including the limiting case where the suspended phase is formed by rigid particles and the mechanism of shear thickening is attributed to the formation of hydroclusters. While considering both contact friction and lubrication hydrodynamics, under some shear conditions, the relative sliding of smooth particles can be inhibited, thus leading to a jamming of the suspension and an enhancement of the shear stress [33,34]. This yield point appears at concentrations that can be significantly below the percolation threshold [35]. It is interesting to observe that the discontinuous transitions, denoting highly non-linear rheological behaviour, have a close relationship with equilibrium phase transitions. In the case of rigid spheres, for example, at low shear rates, light scattering observations suggest that the particles arranged in ordered layers that are lost at higher shear rates. This phenomenon results in a three-dimensional random distribution that has a high degree of interactions among the particles [16,36].

Between the solutions of associating polymers and the suspensions of rigid particles, there is a wide range of intermediate situations that may show shear thickening [37], for example, nanoparticles suspended in a polymer solution [38,39]. If the particles bond physically with the polymer chains and have dimensions smaller than those of the polymer coils, then each chain may be connected to several particles. When the system is in an unperturbed state, in a dilute solution, the most probable configuration is that of isolated chains, and the suspension may flow in a Newtonian manner. In contrast, while applying a shear stress, the changes in polymer conformation may generate new polymer-particle connections, and thus intermolecular bridges and shear thickening [38] may occur. The phenomenon is reversible if, when the shear stress is removed, the chains recover their original random coil configuration.

The question that may arise is how should asphalt and polymer modified asphalts be positioned among the previous examples. This is not easy to answer, since asphalt binders are very complex materials, and their structure is still debated [40]. Nevertheless, they contain all the above-mentioned ingredients. The base asphalts are colloidal suspensions of asphaltene molecules and aggregates stabilized by the resins and dispersed in a maltene medium, in other words, a dispersion or rigid particles in a low molecular weight system. A triblock SBS copolymer has characteristics similar to those of the telechelic polymers, because the external styrene blocks have scarce compatibility with asphalt and show a tendency to agglomerate and expose the central butadiene part to the dispersing medium. Moreover, the combination of polymer and asphaltene clusters dispersed in the maltene matrix has similarities with nanoparticles in a polymer solution. Therefore, it is not surprising that under suitable operating conditions, these materials may be subjected to structural transitions associated with shear thickening [41]. A few examples were presented in a previous work [42].

It is the object of this study to investigate the conditions that favour structural transitions associated with shear thickening in PMAs obtained with SBS triblock copolymers with or without the addition of sulfur and/or polyphosphoric acid (PPA), that strongly influence the internal structure of these binders. Sulfur is the most common vulcanization agent and, in SBS modified asphalt, it is used to promote the formation of chemical bridges between polymer chains. Although sulfur is also able to react with asphalt molecules and can be used as asphalt modifier (sulfur extended asphalts [43]), concentrations used for vulcanization of SBS are very low and these effects can be neglected. PPA is another additive that is able to interact with asphalt. While the mechanism of PPA in asphalt is unknown, the literature discusses the possibility of acid-base neutralization, esterification of deagglomerated asphaltenes, precipitation of asphaltenes with polar insoluble PPA-adducts, or a combination of these mechanisms. The reaction of PPA with asphalt, sometimes referred to as “chemical ageing”, has been widely used to modify the rheological properties of asphalt binders [44].

## 2. Materials and Methods

### 2.1. Materials

The selected straight run asphalt, manufactured by Cenovus Energy (Lloydminster, AB, Canada) with a penetration grade of 200/300 was characterized by Superpave binder specification as PG 52-34. This straight run asphalt with higher penetration (low viscosity) was selected for the following reasons: first, due to feasibility of the use of higher concentrations of modifiers and additives; second, the rheological response of the produced PMAs can be studied at intermediate and high service temperatures without interference of the stiffness introduced by straight run asphalt. Thermoplastic elastomer, a linear tri-block medium molar weight copolymer of styrene and butadiene, Kraton D1101, with average content of styrene of 31 wt.% from Kraton Performance Polymers, Inc. (Houston, TX, USA), was used as the polymer modifier. Technical sulfur (Cenovus Energy, AB, Canada) was used to prepare the crosslinked SBS modified asphalt. Selected crosslinked PMAs were further modified using PPA 115 (containing 83.3% of phosphorus pentoxide) from Innophos Inc.(Cranbury, NE, USA).

### 2.2. Asphalt Modification

Preparation of the PMAs followed the standard blending protocol. A polymer was added to the asphalt binder, heated to a high temperature, and blended until no separate parts of polymer were observed. After this step, dynamic crosslinking with sulfur in the absence of any vulcanization additives took place. The order of adding the components was studied prior to preparing the quaternary PMAs that contained PPA and SBS crosslinked with sulfur. Because the order of adding the components did not show any significant impact on the properties of the product, the technology for preparing the quaternary PMAs was based on adding PPA and then SBS crosslinked with sulfur to straight run asphalt.

### 2.3. Experimental

#### 2.3.1. Shear Viscosity

The prepared PMAs were poured into small containers and stored in a freezer before steady shear viscosity testing. Steady shear viscosity was measured in a strain-controlled rheometer ARES–G2 from TA Instruments (New Castle, DE, USA). The testing temperatures were 50 °C, 60 °C, and 70 °C at shear rates ranging from 10^−4^ s^−1^ to 100 s^−1^. The selection of testing geometries was also considered for steady shear viscosity measurements. Although cone-plate geometries are preferred for viscosity measurements due to the constant shear rate across the geometry, an efflux of asphalts with low viscosity and expulsion/dislocation in case of asphalts with higher viscosity (stronger polymer network) were observed. The selected 25 mm plate-plate geometry with 1 mm gap partially eliminated this issue and thus was selected for all steady shear viscosity measurements. To eliminate the mechanical history of the sample, a small amount of asphalt was heated in an oven at 165 °C for 10 min and then directly poured on the tested geometry. The excess material was trimmed. The prepared asphalt sample was left for 30 min to achieve temperature equilibrium and allow for the relaxation of normal forces. A fresh sample was always used to take steady shear viscosity measurements.

#### 2.3.2. Cryo-SEM

A scanning electron microscope JEOL 7500 F (Jeol Ltd., Tokyo, Japan), equipped with a field emission gun and cryogenic mode system from Quorum, was used to investigate the internal structure of the PMAs. A small sample of asphalt, heated to 165 °C for 10 min, was directly transferred to a 3 mm long copper rivet, which was placed to slushed liquid nitrogen. A supercooled sample was transferred in a vacuum to a preparation chamber, cooled to −140 °C, and then fractured using a front-mounted scalpel-blade probe. To eliminate potential fine water slivers generated during fracturing, the sample was exposed to −90 °C for 10 min. After this stage, the sample was coated with a thin layer of Pt/Pd plasma for 60 s. The morphology of the prepared specimens was observed at −175 °C using the gentle beam mode and a working distance of 6 mm.

## 3. Results and Discussion

### 3.1. Base Asphalt

Starting from the base asphalt, Figure 1 shows the viscosity functions at three different temperatures. The material exhibits a zero-shear viscosity ($\eta_{0}$) that corresponds to classical Newtonian behaviour was observed for a few decades of shear rate, until a sharp shear thinning occurs. The latter corresponds to the disruption of the colloidal equilibrium, with disaggregation and possible orientation of the asphaltene clusters. The relative flow of the molecules is determined by their mutual interactions, which may be broken by an increase in temperature and/or mechanical stress. Therefore, $\eta_{0}$ is getting smaller with temperature and, at a constant temperature, shear thinning is induced by the mechanical stress. Low viscosity means reduced friction, and thus, at higher temperatures, shear thinning shifts to the right. However, if the same curves are plotted as a function of shear stress, not reported here, then the reduction in viscosity occurs approximately at the same value irrespective of the temperature. None of the tested operating conditions showed shear thickening, and therefore, in the base asphalt, the colloidal suspension did not show aggregation induced by an external stress.

### 3.2. Base Asphalt and PPA

Shear viscosity was also tested using the base asphalt modified with increasing quantities of PPA. At low PPA concentrations, the qualitative behaviour of the viscosity functions was very similar to that of the unmodified binder. The main difference with respect to the base asphalt was a stiffening effect manifested by higher values of $\eta_{0}$ if compared at the same temperature. This result was not surprising because PPA transformed part of the naphthene and polar aromatics into asphaltenes, thus increasing the concentration of polar and high molecular weight molecules. At increasing PPA concentrations, the stiffening effect became more and more evident (about one order of magnitude for the higher concentration), but there was also variation in the shape of the curves (Figure 2). The Newtonian plateau became shorter and a gradual decrease in viscosity preceded the sudden final drop that happened at the same shear rates observed for the base asphalt. This behaviour was best observed at 50 °C and suggested that the newly formed associated structures had a strength distribution that corresponded to different shear rates needed for their breakage. An increase in temperature caused a pre-breakage of the weakest bonds, and thus the descending behaviour was less evident but still observable at 60 °C and 70 °C.

### 3.3. Base Asphalt and SBS

The functions $\eta \left(\dot{\gamma }\right)$ for an asphalt blend modified with 2, 3, 4, and 5% SBS D1101 copolymer were recorded at temperatures ranging from 30 °C to 60 °C. As shown in Figure 3, the curves increased in 𝜂_0_ that paralleled the polymer concentration. The lower polymer content behaved similarly to that of the base asphalt (Figure 3). At temperatures equal or below 50 °C, there was a long zero-shear viscosity plateau followed by a shear thinning region. However, the latter followed a less steep path. Compared to the unmodified asphalt, the presence of a low concentration of dispersed swollen polymer particles resulted in a decrease in viscosity over a wider range of shear rates. This qualitative trend disappeared at 60 °C, where we observed the first mild shear thickening that produced a shoulder in the viscosity function, until the usual sharp shear thinning occurred.

The range of possible mechanisms leading to shear thickening were listed in the Introduction section and will be further discussed below. However, a premise is necessary before commenting on these curves. In a recent paper, Wagner et al. observed that the viscosity overshoot at very low shear rates in diluted polymer solutions can be an artefact attributable to an inappropriate sampling interval [45]. If the latter was too short, then the sample did not reach the steady state condition and the estimated shear viscosity was wrong. The observation was made for diluted polymer solutions but may apply also to other systems. The authors reported the transient behaviour of the stress growth in experiments performed at fixed values of the shear rate (${\dot{\gamma }}_{0}$). Depending on the shear rate, after a period corresponding to the sampling time, the shear stress ($\tau^{+}=f\left({\dot{\gamma }}_{0}\right)$) resulted in values either lower, equal, or higher than its final steady state value. If the measurement was taken during this transient period, then the calculation of the shear viscosity did not reflect its real value. At very low shear rates it was underestimated, at intermediate shear rates it was overestimated, and only at high shear rates was the steady state reached, thus giving the correct value of $\eta_{0}$. To overcome this issue, the authors suggest an optimum acquisition time $dt\;=\;60+\frac{5}{\dot{\gamma }}$ (expressed in seconds).

It must be emphasized that the problem occurred using diluted polymer solutions that have viscosities much lower than those of asphalt binders at the temperatures tested in this work. In our case, high viscosity allows for testing at very low shear rates, and starting around 10^−3^ s^−1^ would mean an acquisition time over 5000 s per measurement of one point, which is not reasonable. Therefore, we paid attention to the removal of the mechanical history of the sample and the reproducibility of the data. However, it must be stressed that the shapes of our curves are different from those observed in the presence of the above-described artefact, where the viscosity function starts rising, reaches a maximum, and then goes down to $\eta_{0}$. If the sample has not reached the steady state during steady shear viscosity measurements, the curve cannot show a plateau, like in our data, before the viscosity overshoot. Therefore, we believe that shear thickenings reported in our work are not artefacts but are rather due to alterations of the internal structure.

At 3 wt.% of polymer loading, not reported here, the behaviour is similar to the case of 2% SBS, but at lower temperatures, the region of supposed Newtonian plateau denotes a moderate tendency of shear thinning. At 4 and 5 wt.% of SBS content, all effects become more evident (Figure 4). The shear thickening is visible at lower temperatures and starts at very low shear rates, so that the initial Newtonian plateau is no longer visible. Then, there is a long region of moderate shear thinning that preludes the final, sharp one. Moreover, at higher temperatures, shear thickening is more pronounced, and the material tends to show the second plateau before the final drop in viscosity. Therefore, we can summarize the effect of increasing polymer content on the viscosity function as follows:The appearance of a shear thickening region that, at higher temperatures, is more pronounced and shifts to the left in the shear rate axis.The presence of a gradual shear thinning preceding the sharp final one.

The above-described shear thickenings can be interpreted coherently using the mechanisms described in the Introduction section. The behaviour of the block copolymer dispersed in the asphalt binder is similar to that of associating polymers in solution. When both the polymer content and temperature are low, the rheological behaviour is predominantly determined by the asphalt matrix, and thus the viscosity curves resemble those of the base asphalt. Nevertheless, the presence of the polymer is not negligible as shown by both $\eta_{0}$ (almost doubled at 50 °C by a 2% polymer loading) and the different shape of the shear thinning region. Both differences indicate that even a small amount of SBS has a significant effect on the rheological behaviour of the binder. 

The smooth shear thinning region can be easily interpreted based on the polymer characteristics. As a general rule, polymeric materials do not show sharp and well-defined transitions. For example, different from the case of low molecular weight substances, in polymeric materials, melting or crystallization are not isothermal because the macromolecules have different lengths and degrees of branching. Both factors affect the ability of polymeric materials to move and rearrange their conformation. In other words, each macromolecule has its peculiar characteristics, and the molecular weight distribution results in a distribution of melting or crystallization temperatures. Analogous considerations are valid for the associative behaviour of the chains [46]. In asphalt modification, the bridges and bonds responsible for the polymeric network and the polymer-asphalt interactions have a distribution of lengths and strengths, respectively. A short polystyrene segment is easily detachable from the polystyrene domain. In addition, a short polybutadiene segment may bond with a limited number of asphalt molecules. Therefore, a gradual increase in shear rate during the viscosity test results in a gradual breaking of the physical bonds, starting with the weakest ones, which is why the structural transitions leading to both shear thickening and shear thinning are spread over a wide range of shear rates. At low shear rates, shear stress induces the formation of new bridges between the polystyrene domains, as it happens in solutions of telechelic polymers, while at high shear rates, breaking prevails. Further, if temperature and polymer loading are increased, the contribution of the polymeric part to the whole rheological behaviour becomes more important, thus leading to more pronounced non-linear behaviour, as shown in Figure 4.

### 3.4. Base Asphalt, SBS and Sulfur

In the case of the modified Cenovus Energy 200/300 Pen grade straight run asphalt, it was observed that transfer of elastomeric properties, which altered the deformation-relaxation processes, occurred at a minimum of 3 wt.% of SBS. This PMA was further crosslinked with different amounts of sulfur ranging from 0.04 wt.% to 0.20 wt.%. Viscosity functions for asphalt blends modified by 3 wt.% SBS D1101 and different quantities of sulfur are shown in Figure 5. Generally, a higher content of sulfur means a higher degree of crosslinking and gives an effect similar to what is observed for an increase in polymer concentration: higher magnitudes of shear viscosity and a more pronounced shear thickening. For example, Figure 5 shows a comparison of the viscosity functions recorded at 60 °C. The main effect of sulfur lies in the formation of chemical bonds among the polybutadiene segments. To a small extent, sulfur could be also attached to asphalt molecules, but given the very small concentrations of sulfur used in commercial vulcanization, this effect might be considered insignificant. Therefore, sulfur promotes the formation of polymer networks through the formation of predominantly polysulfide bonds. A formed polymer network in PMAs practically eliminates issues of separation of polymer from asphalt, for example, during storage at high temperatures, but it also results in a transfer of elastomeric properties to the whole asphalt blend, and thus significantly enhances the thermo-rheological properties.

As shown in Figure 5, at low concentrations of sulfur, there is limited chemical crosslinking, and the properties of such PMAs are dominated by the physical network of ultra-fine, dispersed non-crosslinked polymer particles. At higher concentrations of sulfur, the departure from Newtonian behaviour is more apparent, and the higher “viscosity overshoot” reflects an increased resistance of PMAs to deformation. In non-crosslinked SBS D1101 asphalt blends, shear thickening was explained by the formation of a transient physical network, and a similar mechanism should also be valid in this case. However, the question is how can sulfur promote transient networks if it creates chemical bonds among polymeric molecules that are supposed to survive at moderate thermo-mechanical stresses? A look at the whole modification process may help to answer this question. During the preparation of the blends, there is a preliminary blending of binder and polymer, and then sulfur is introduced in the system. This blending sequence is necessary, because the introduction of sulfur too early may freeze the movements of the polymer chains before the desired morphology can fully develop. Moreover, it is well known that sulfur has an excellent ability to improve the storage stability of PMAs. Therefore, providing stable linkages, sulfur has a double effect: first, it acts as a reinforcing agent that increases the strength and elasticity of the polymeric network; second, it limits the tendency to phase separation that occurs when the blend is still at high temperature but no longer under blending conditions. The chemical links limit the number of chains able to recoil at the end of blending when the temperature is still high, and no shear is applied. Therefore, without sulfur, the macromolecular chains are forced into a thermodynamic unstable conformation. This metastable state involves the whole polymer, and thus the material has a higher network density, internal stress, and number of chain-ends prone to conformational rearrangements. In other words, although small concentrations of sulfur can generate only a limited number of chemical bonds, the resulting alteration of the PMA structure significantly affects the physical connections of the network. Therefore, the increase in viscosity due to the formation of chemical bonds is only mild, while the properties of the physical, non-crosslinked network still prevail. This result can be seen in Figure 6, where the shear stress is portrayed as a function of shear rate; Figure 6 shows that the over-stress present in the sulfur modified binders disappears at high shear rates, thus confirming its transient nature. In the new configuration, more chain-ends are able to jump from a styrene domain to another one, and thus, a more pronounced overshoot occurs. Moreover, at shear rates around 1 s^−1^, the number of new elements contributing to the network is definitely overruled by those leaving it, shear thinning starts, and the stress curves merge to values similar to those of the binder without sulfur. This situation confirms that sulfur is a major contributor in the structural conformation of the chains.

The cryo-SEM images shown in Figure 7 indicate how much sulfur may impact binder morphology. The binder without the chemical crosslinker shows isolated polymer-rich islands highlighted by a white ellipse in Figure 7a. At higher magnification (Figure 7b), the islands contain a mesh with boundaries composed of small bright dots. The latter are the polystyrene domains, having dimensions in the order of tens of nanometers. After adding sulfur, a polymer-rich phase is no longer distinguishable, and the above-mentioned polystyrene domains appear uniformly dispersed in the asphalt matrix. Some of such domains are indicated with an arrow in (Figure 7c).

### 3.5. Base Asphalt, SBS, Sulfur and PPA

The shear viscosity of asphalt blends modified by 3 wt.% SBS D1101, 0.12 wt.% sulfur and PPA is quite different from the previous blends. Again, data were recorded at different temperatures and PPA concentrations. Figure 8 shows data at 60 °C and variable PPA content as a function of the shear stress. 𝜂_0_ does not increase monotonically with PPA content. At the same time, the viscosity overshoot disappears, and the curves show a double-step shear thinning. Both differences indicate a strong interference between PPA and the polymer network, which seems to be inhibited by the acid. Therefore, the easiest interpretation is that the SBS network is still present but with a strength and extension comparable to that obtained with a lower polymer content in the absence of PPA. The mechanical behaviour is the result of the comparable contribution of the SBS and PPA. Therefore, the first shear thinning region is caused by the deformation and rearrangement of the sulfur crosslinked polymer network. Then, at shear stresses around 10^−4^ MPa, the shear viscosity is dominated by the binder, and the second decrease is related to the destruction of the internal structure of the base asphalt as modified by the PPA. In fact, as PPA content increases, the second shear thinning region becomes more important, and this region remains the only one observable at higher PPA contents.

Generally, at small concentrations, PPA can be used as a sole modifier of straight run asphalts. The alteration of the colloidal system of asphalt by acid enhances resistance to permanent deformation without major changes in low temperature behaviour. According to several authors, PPA may act as a crosslinker for SBS-modified asphalts. This conclusion was based on the observation that introducing PPA to non-crosslinked SBS modified asphalt improved the delayed elastic response compared to the effects of modifiers alone. From this perspective, one could speculate that when PPA alters a colloidal system of asphalt that has the capability to crosslink SBS, then it should be able to enhance relatively weak 3-D polymer networks created by sulfur significantly. If this mechanism occurred, the polymer network would strengthen due to the higher crosslink density, which could result in a higher viscosity overshoot. In contrast, the results presented in our work indicate the appearance of double-step shear thinning regions at the expense of viscosity overshoot. It is generally accepted that the effect of a polymer, and thus also the effect of a polymer network in an asphalt binder with lower penetration (higher viscosity), is less pronounced. This effect could be attributed to the higher stiffness of the asphalt binder, which may “shield” the features of modifiers. Although the stiffening effect in the case of PPA is different compared to distillation processes or to air-blowing technology, the higher stiffness may have a similar effect on the structural response of PMAs in steady shear viscosity measurements.

## 4. Conclusions

Modified asphalts are important construction materials. In order to predict the performance of these materials, the internal structure needs to be thoroughly understood. Rheological tests, particularly steady shear viscosity measurements, have been used for the investigation of different responses of modified asphalts. 

While straight run asphalts have been found to shift from Newtonian behaviour at low shear rates to a sharp shear thinning region, the addition of modifiers can have a profound effect on both the shape and magnitude of the viscosity functions. In terms of microstructure, adding PPA altered the colloidal system of asphalt by transforming naphthene and polar aromatics into asphaltenes. The newly formed asphaltene structures have a strength distribution that leads to a shear thinning region spread over a wider range of shear rates. In contrast, a shear thickening region was observed at high concentrations of SBS. This could be explained by a gradual breakage and rearrangement of the bridges formed by butadiene segments in a way that resembles the solutions of telechelic polymers. The addition of sulfur, as a crosslinking agent, to SBS-modified asphalts resulted in the formation of chemical bridges between polybutadiene chains. These affected the conformation of the polymer network and also led to pronounced shear thickening phenomena. Finally, the presence of PPA in vulcanized PMAs overruled the transitions in the polymer network, thus eliminating the viscosity overshoot and leading to a multi-step shear thinning region.

The reported results demonstrate that viscosity curves could serve as a useful tool for investigation of the role of modifiers in asphalt binders. The wide range of shear rates covered by the tests allow for the study of structural rearrangements in both the linear and nonlinear viscoelastic regions. The response of modified asphalts in shear viscosity measurements correlated with direct observation of internal structure via cryo-SEM. From a practical point of view, a proper interpretation of shear viscosity functions could be useful for understanding the role of single modifiers, with potential optimization of their concentrations when formulating binary, ternary, or quaternary asphalt blends.

## Figures and Tables

**Figure 1 materials-15-07551-f001:**
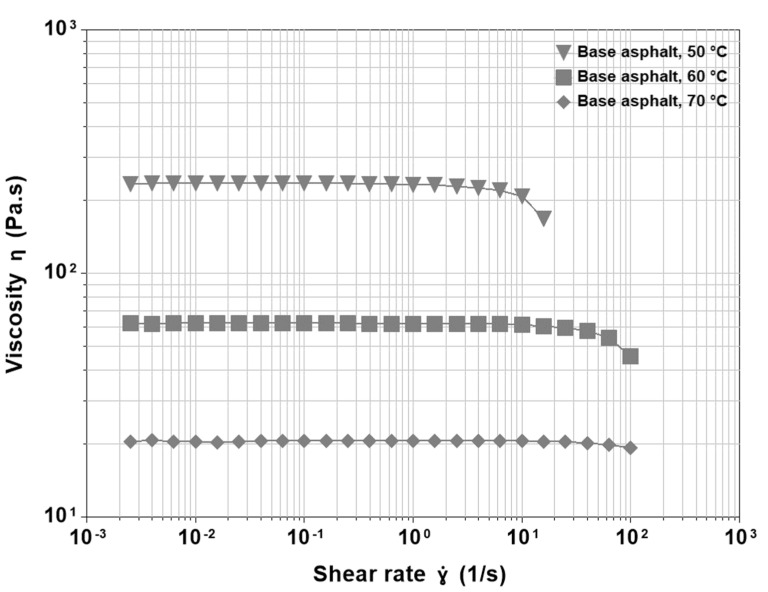
Viscosity functions at different temperatures for Cenovus Energy 200/300 Pen grade straight run asphalt.

**Figure 2 materials-15-07551-f002:**
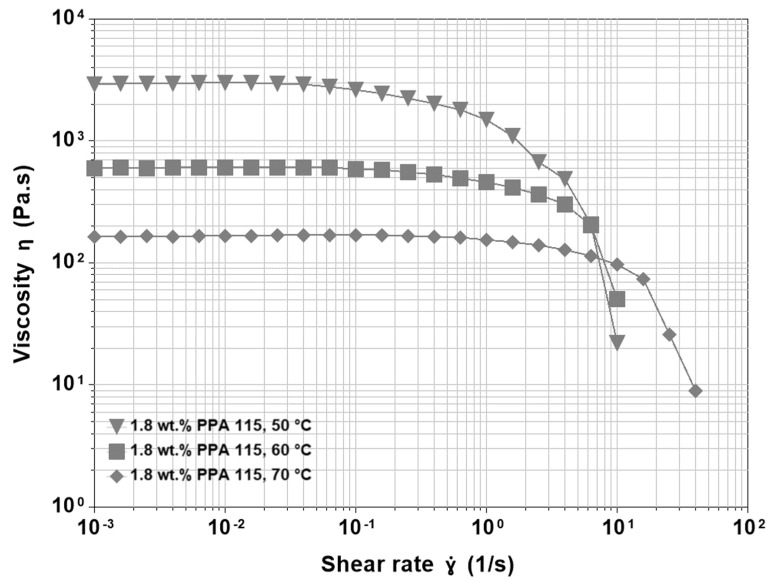
Viscosity functions at different temperatures for 200/300 Pen grade modified by 1.8 wt.% PPA 115.

**Figure 3 materials-15-07551-f003:**
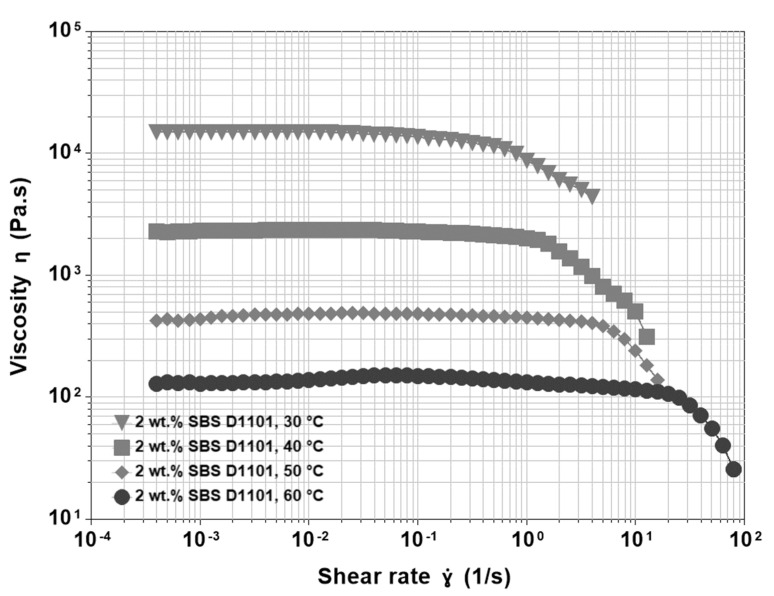
Viscosity functions at different temperatures for 200/300 Pen grade modified by 2 wt.% SBS D1101.

**Figure 4 materials-15-07551-f004:**
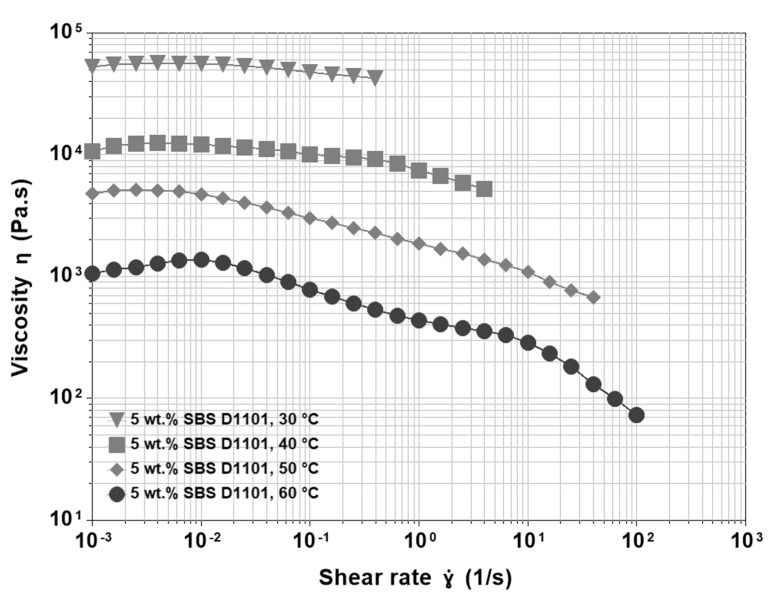
Viscosity functions at different temperatures for 200/300 Pen grade modified by 5 wt.% SBS D1101.

**Figure 5 materials-15-07551-f005:**
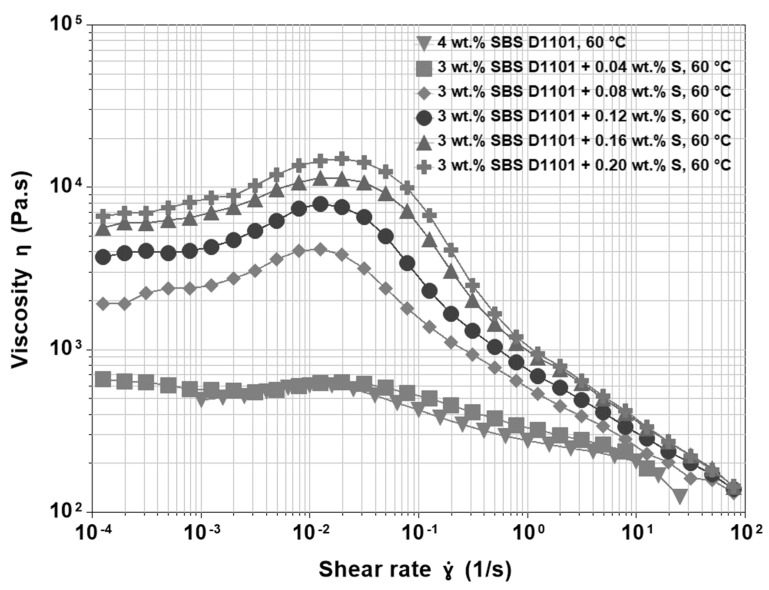
Viscosity functions at 60 °C for 200/300 Pen grade modified by 3 wt.% SBS D1101 crosslinked with sulfur ranging from 0.04 wt.% to 0.20 wt.% compared to 200/300 Pen grade modified by 4 wt.% SBS D1101 without sulfur.

**Figure 6 materials-15-07551-f006:**
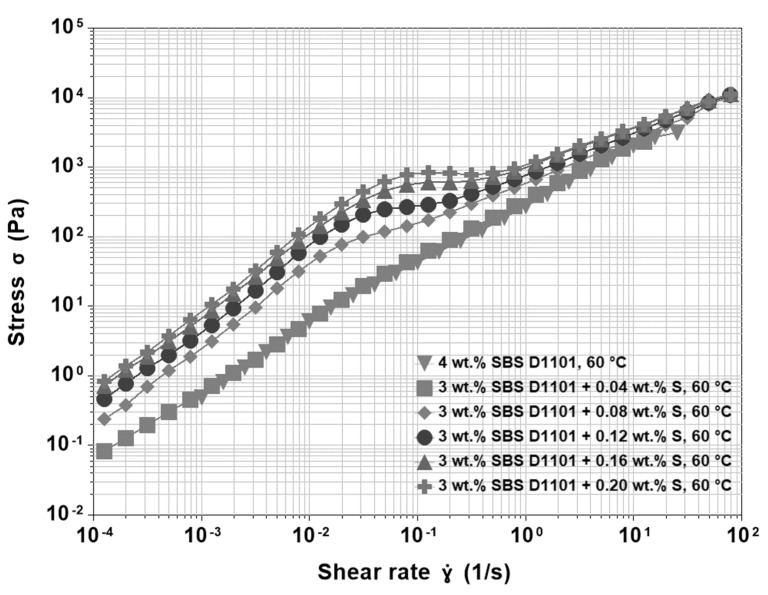
Shear stress as a function of shear rate at 60 °C for 200/300 Pen grade modified by 3 wt.% SBS D1101 crosslinked with sulfur ranging from 0.04 wt.% to 0.20 wt.% compared to 200/300 Pen grade modified by 4 wt.% SBS D1101 without sulfur.

**Figure 7 materials-15-07551-f007:**
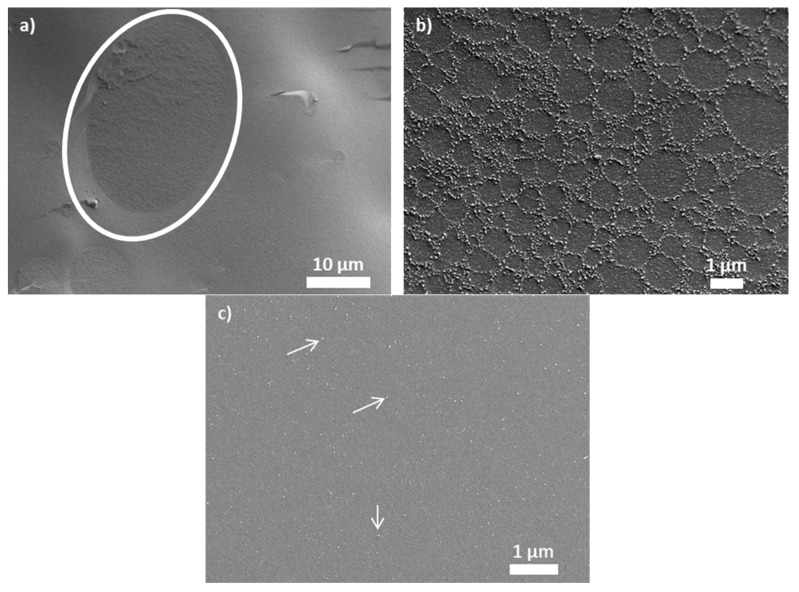
Morphology of asphalt blends obtained using Cryo-SEM: (**a**) 3.0 wt.% SBS D1101 magnified ×2000; (**b**) 3.0 wt.% SBS D1101 magnified ×10,000; (**c**) 3.0 wt.% SBS D1101 with 0.12 wt.% sulfur magnified ×15,000.

**Figure 8 materials-15-07551-f008:**
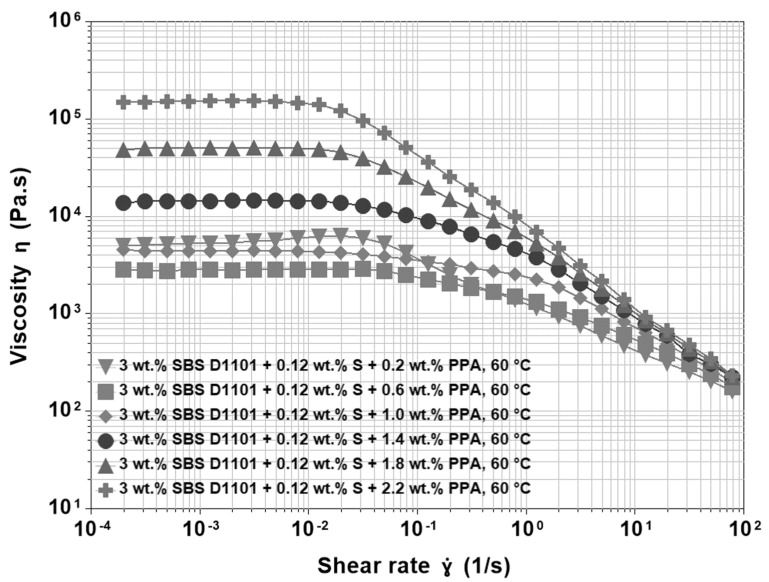
Viscosity functions at 60 °C for 200/300 Pen grade modified by 3 wt.% SBS D1101 crosslinked with 0.12 wt.% sulfur and PPA ranging from 0.2 wt.% to 2.2 wt.%.

## Data Availability

All the data are available within the manuscript.

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
