# Peer review of "Shear Viscosity Overshoots in Polymer Modified Asphalts"

_materials, 2022, doi:10.3390/ma15217551_

Round 1
Reviewer 2 Report
Shear viscosity overshoots in polymer modified asphalts
The article investigate the conditions that favour structural tran-sitions associated with shear thickening in PMAs obtained with SBS triblock copolymers with or without the addition of sulfur and/or polyphosphoric acid (PPA), that strongly influence the internal structure of these binders. Sulfur is the most common vulcanization agent and, in SBS modified asphalt, it is used to promote the formation of chemical bridges between polymer chains.
This study is very interesting. Few comments are given below to further improve the quality of the article.
Abstract need revision with some quantitative results.
Some more latest studies are required in the introduction section to further highlight the importance of this study.
Sanchana, I. C., Sandeep, I. J. S., Divya, P. S., Padmarekha, A., & Murali Krishnan, J. (2022). Determination of linearity limit of bitumen and mastic using large-amplitude oscillatory shear. International Journal of Pavement Engineering, 1-15.
Shalabi, F. I., Mazher, J., Khan, K., Alsuliman, M., Almustafa, I., Mahmoud, W., & Alomran, N. (2019). Cement-stabilized waste sand as sustainable construction materials for foundations and highway roads. Materials, 12(4), 600.
Genreally, the quality of graphs is very poor.
Authors must summarized results in more systematic way with reference to the previous studies.
Please further shorten the conclusions.
Round 2
Reviewer 1 Report
Only one comment,
I suggest the authors to replace the old references with modern references. one of the references is more than 40y old. [26, 24, 13, 5, and 4]
My regards...
Author Response
References 13, 24 and 26 were replaced. References 4 and 5 were maintained since the papers by Macosko are sort of milestone and deserve to be cited. Nevertheless, other more recent references were added.